# New Roles of bZIP-Containing Membrane-Bound Transcription Factors in Chromatin Tethering and Karyoptosis

**DOI:** 10.3390/ijms262210896

**Published:** 2025-11-10

**Authors:** Dohyun Jeung, Xianzhe Li, Yong-Yeon Cho

**Affiliations:** BK21-4th, College of Pharmacy, The Catholic University of Korea, 43, Jibong-ro, Wonmi-gu, Bucheon-si 14662, Gyeonggi-do, Republic of Korea; jungdo357@gmail.com (D.J.); a111lixianzhe@gmail.com (X.L.)

**Keywords:** type II membrane-bound transcription factor, nucleoskeleton, nuclear integrity, regulated cell death

## Abstract

The nuclear membrane has emerged as a dynamic regulatory platform coordinating genome organization, mechanotransduction, and regulated cell death (RCD). Beyond its barrier function, the nuclear skeleton—comprising lamins, actin–myosin isoforms, nuclear matrix proteins, and the LINC complex—supports nuclear integrity and gene regulation. Recent evidence shows that type II membrane-bound bZIP transcription factors such as cAMP-responsive element-binding protein 3 (CREB3) and CREB3L1 localize to the inner nuclear membrane (INM), linking chromatin tethering with stress signaling. Their stress-induced cleavage by S1P/S2P disrupts chromatin anchoring and, in some contexts, triggers karyoptosis, a novel form of RCD defined by nuclear rupture. These findings position the nuclear envelope (NE) as a mechanosensitive signaling hub with direct implications for disease and therapy. In this review, we provide a comprehensive discussion on how type II membrane-bound bZIP transcription factors and chromatin acting as a nucleoskeleton cooperate to regulate nuclear membrane integrity.

## 1. Introduction

The nucleus is a highly compartmentalized organelle that not only safeguards the genome but also orchestrates replication, transcription, and chromatin organization. The nuclear envelope (NE), composed of the outer and inner nuclear membranes (ONM and INM), nuclear pore complexes (NPCs), and the underlying nuclear lamina, provides the structural and functional foundation for nuclear homeostasis. Traditionally viewed as a passive barrier, the NE is now recognized as a dynamic and mechanosensitive signaling hub that integrates nuclear architecture with gene regulation and cell fate [1].

A critical determinant of nuclear stability is the NE and its associated protein complexes, which together form a dynamic architectural network rather than a rigid internal skeleton. This network includes lamins, nuclear actin–myosin isoforms, matrix proteins such as SAF-A and SATB1, the LINC complex that links the nucleus to the cytoskeleton, and chromatin itself [2]. Collectively, these elements preserve nuclear shape and elasticity, anchor heterochromatin at the periphery, and transmit mechanical cues from the cytoplasm to the genome. Disruption of this integrated system compromises nuclear integrity and has been implicated in laminopathies, cancer, and premature aging syndromes [3]. Thus, maintaining nuclear membrane integrity is essential not only for protecting the genome but also for ensuring appropriate responses to physiological and pathological stress.

Recent findings have expanded this framework by implicating type II membrane-bound bZIP transcription factors, notably cAMP-responsive element-binding protein 3 (CREB3) and ATF6, in nuclear membrane biology [1]. These proteins, originally described as ER- and Golgi-resident stress sensors [4,5], can also localize to the INM [1,4,6]. In their full-length form, they act as chromatin tethers, physically linking the nuclear lamina to peripheral heterochromatin [1,6]. This tethering is crucial for counterbalancing the outward pressure of compacted chromatin with the inward tension provided by lamins and the cytoskeleton [1].

Importantly, the stress-induced cleavage of full-length form of CREB3 (called CREB3-FL) by site-1 protease (S1P) and site-2 protease (S2P) has been observed at the INM [7]. This suggests that S2P-mediated intramembrane proteolysis liberates the N-terminal bZIP domain, which may act as a direct source of the cleaved form of CREB3 (called CREB3-CF) to regulate stress-adaptive gene expression in the nucleus. However, when this cleavage is dysregulated, the cleavage simultaneously disrupts chromatin anchoring, thereby weakening nuclear integrity [1]. In extreme cases caused by strong UVB irradiation, this dysregulation leads to NE rupture, chromatin leakage, and the initiation of karyoptosis, a newly described form of regulated cell death (RCD) [1]. Recent discoveries propose that karyoptosis, a form of RCD initiated by the mechanical collapse of the NE and characterized by chromatin leakage, is mechanistically distinct from other nuclear rupture–associated cell death modalities such as apoptosis, necroptosis, or ferroptosis. This concept has been supported by the comparative analyses of UVB-induced cell death pathways [1], emphasizing karyoptosis as a transcription factor–dependent process coupled with the loss of chromatin tethering and nuclear disassembly. A brief comparison of key RCD types is provided herein to guide the reader. Such findings highlight a dual role for INM-anchored transcription factors: maintaining nuclear stability in their uncleaved state and mediating stress-responsive transcriptional programs upon proteolysis.

Understanding the mechanisms that regulate nuclear membrane integrity—how structural scaffolds, chromatin tethering, and membrane-bound transcription factors coordinate—remains a major frontier in nuclear biology. Elucidating these processes will not only clarify the fundamental principles of nuclear organization but also shed light on how their dysregulation contributes to disease pathogenesis and RCD.

## 2. Nuclear Membrane Constituents

The NE is a dynamic and specialized double-membrane structure that defines the nuclear boundary and maintains genomic compartmentalization [8]. It comprises the NPCs, the ONM and INM, and the underlying nuclear lamina. These membranes harbor distinct sets of integral membrane proteins that play essential roles in nuclear architecture, chromatin organization, signal transduction, and cell fate regulation [9]. The constituents in the NE (Figure 1) not only safeguard the genome from cytoplasmic insults but also mediate critical processes including genome organization, mechanotransduction, and RCD such as karyoptosis [1,6].

### 2.1. Nuclear Pore Complexes

The NPCs are the exclusive gateways between the nucleus and cytoplasm. Each NPC comprises ~30 nucleoporins (NUPs) and assembled into an octagonal scaffold that regulates passive diffusion and active, receptor-mediated transport of macromolecules [10,11,12,13]. FG-nucleoporins lining the central channel provide a selective permeability barrier that allows transport receptors to shuttle cargo in a regulated, high-throughput manner [12,14].

Beyond transport, nucleoporins such as Nup153, Nup98, and TPR associate with chromatin and regulate transcription at the nuclear periphery [15,16,17]. NPC-embedded proteins like POM121 and NDC1 stabilize pore-membrane junctions during mitosis and mechanical stress [18]. Integral INM proteins are targeted via the diffusion–retention model: they insert into the ER, diffuse laterally into the ONM, and cross the pore membrane [19]. Smaller proteins (<~70 kDa) enter via the peripheral NPC channels, whereas larger proteins require NLS motifs and import receptors [20]. The disruption of NPC integrity impairs nuclear homeostasis and can contribute to RCD, including karyoptosis [6,21].

### 2.2. Integral Proteins in the ONM

The ONM, continuous with the ER, contains specialized integral proteins for cytoskeletal anchoring and nuclear positioning. The key ONM proteins are nesprins, which are giant spectrin-repeat proteins linking actin filaments through their N-terminal domains, while their C-terminal KASH domains interact with SUN proteins in the INM to form the LINC complex [22]. Nesprin-2 additionally connects to intermediate filaments via plectin, ensuring nuclear positioning and mechanical stability [23,24].

In muscle cells, the dystrophin–glycoprotein complex connects the cytoskeleton to the ONM, contributing to mechanotransduction. Through these linkages, the ONM integrates cytoskeletal forces with nuclear structure and gene expression [25]. While type II membrane-bound transcription factors are enriched in the INM, their potential interactions with cytoskeletal elements at the ONM remain largely unexplored.

### 2.3. Integral Proteins in the INM

The INM contains proteins critical for nuclear architecture and chromatin organization. Examples include LBR, which binds lamin B and HP1 [26,27]; LEM-domain proteins (LAP2β, emerin, MAN1) that tether DNA via BAF [28,29]; SUN1/2, which link to ONM nesprins [23]; TMEM43, associated with cardiomyopathy [30]; and PRR14, which anchors HP1-marked chromatin to lamins [31].

Notably, CREB3 has been identified at the INM, where its N-terminal domain binds lamins and chromatin [7], suggesting a dual role as a structural tether and transcriptional regulator. The precise mechanisms of CREB3 targeting and cleavage at the INM remain to be clarified, but its involvement highlights a regulatory axis linking nuclear stability and cell fate.

### 2.4. INM-Localized Integral Proteins Interacting with Chromatin

Several INM proteins directly tether chromatin to the nuclear periphery, thereby stabilizing nuclear shape and maintaining heterochromatin organization. LBR recognizes H3K9-methylated histones [32,33], LAP2β recruits HDAC3 to form transcriptionally repressive domains [34,35], and MAN1 modulates TGF-β signaling through interactions with Smad2/3 [36]. PRR14 dynamically anchors HP1-bound chromatin to lamin A in a cell cycle–dependent manner [31].

A notable addition is CREB3, a type II bZIP transcription factor whose nucleoplasmic N-terminal domain binds both DNA and lamins [7]. This tethering function helps to balance the outward pressure generated by chromatin compaction with the inward tension imposed by the nuclear lamina [1,7,37]. Under stress conditions, the cleavage of CREB3-FL disrupts these attachments, leading to nuclear instability and, when the stress is excessive, nuclear rupture and karyoptosis [7].

Despite these insights, the detailed molecular mechanisms pertaining to nuclear membrane integrity remain poorly understood. How CREB3 directly interacts with chromatin and DNA, how its proteolytic cleavage is regulated at the INM, and how the full-length protein is specifically localized to the INM are still unanswered questions. Although the roles of bZIP transcription factors have been classically studied, their structural features and the potential for diverse homo- and hetero-dimerization events suggest a novel framework in which such transcription factors may tether specific genomic regions to the nuclear membrane, thereby integrating structural stability with transcriptional control.

## 3. bZIP Family Members

### 3.1. Background of bZIP Transcription Factors

The bZIP superfamily is one of the largest and most well-characterized groups of transcription factors in eukaryotes. bZIP proteins are defined by the presence of a highly conserved bZIP domain, which consists of two distinct regions: a basic region (BR) involved in DNA binding and an LZ motif that mediates dimerization [38]. The basic region typically contains clusters of basic amino acids, such as arginine and lysine, that facilitate DNA binding. A consensus motif for this region is N-x7-R-x9-L-x6-L-x6-L, which enables the recognition of specific DNA sequences, usually palindromic or pseudo-palindromic motifs in the major groove of DNA [39]. These DNA motifs often contain the canonical ACGT core, such as the cAMP response element (CRE: TGACGTCA) or the TPA response element (TRE: TGAGTCA) [40,41]. The LZ region consists of heptad repeats in which leucine or other hydrophobic residues appear at every seventh position, forming an amphipathic α-helix. This structure promotes homo- or heterodimerization between bZIP monomers, a prerequisite for DNA binding [42]. Through dimerization, bZIP proteins expand their DNA recognition repertoire and modulate transcriptional specificity [43,44]. Functionally, basic LZ (bZIP) transcription factors act as molecular sensors that integrate various extracellular and intracellular signals to regulate gene expression. They control a wide range of biological processes including cell proliferation, differentiation, apoptosis, stress responses, metabolism, and circadian rhythm [1]. Their ability to rapidly respond to stimuli such as oxidative stress, ER stress, and hormonal signals makes them key nodes in transcriptional regulatory networks. In humans, 55 bZIP transcription factors have been identified and are classified into several subfamilies based on sequence homology and DNA-binding preferences based on the Uniprot data base (accessed on 1 October 2025; https://www.uniprot.org/uniprotkb?query=%28family%3A%22bZIP+family%22%29&facets=reviewed%3Atrue%2Cmodel_organism%3A9606) (Figure 2). These include AP-1 (Activating Protein-1), which includes the c-Fos, c-Jun, and ATF proteins involved in cell growth and stress responses; C/EBP (CCAAT/enhancer-binding proteins), which regulates immune function, adipogenesis, and hematopoiesis; ATF/CREB (Activating Transcription Factor/cAMP-Responsive Element-Binding), which is the key mediators of stress signaling and unfolded protein response; musculoaponeurotic fibrosarcoma oncogene family (MAF), which is involved in development and oncogenesis; Cap ‘n’ Collar (CNC), which includes NRF1 and NRF2, which play major roles in oxidative stress defense; and PAR (proline and acidic amino acid-rich), which is involved in circadian rhythm and metabolic regulation [45,46,47]. Notably, bZIP transcription factors often undergo post-translational modifications—including phosphorylation, acetylation, and proteolytic cleavage—that modulate their DNA-binding activity, nuclear localization, and transcriptional output. This is particularly important for stress-responsive bZIPs such as ATF6 and CREB3, which are synthesized as membrane-bound precursors and become transcriptionally active only upon regulated intramembrane proteolysis (RIP) [7,48,49,50]. Furthermore, bZIP factors frequently form heterodimers across subfamilies, enabling combinatorial diversity in gene regulation [47]. This dimeric flexibility underlies their context-dependent roles in normal physiology and disease, including inflammation, cancer, and neurodegeneration.

### 3.2. bZIP Family Members Lacking Transmembrane Domain

Among the bZIP superfamily, most transcription factors lack TM domains and function as soluble nuclear proteins (Figure 2). These factors play essential roles in proliferation, differentiation, apoptosis, metabolism, circadian rhythm, and stress responses. Despite their functional diversity, they share a common mode of DNA recognition: binding to palindromic or CRE/TRE-like motifs containing the canonical ACGT core sequence [40,41]. Each subfamily exhibits characteristic consensus motifs.

The AP-1 subfamily (Jun, Fos, and ATF dimers) binds the TPA response element (TRE: 5′-TGA(C/G)TCA-3′) and, in association with ATF proteins [51,52], the CRE motif [53]. AP-1 activity is primarily regulated by MAPK-dependent phosphorylation and integrates signals controlling proliferation, apoptosis, and inflammation [54]. While c-Jun and c-Fos act as oncogenic drivers [55], JunB functions as a tumor suppressor [51,52], illustrating the context-dependent duality of AP-1.

The PAR subfamily (DBP, TEF, and HLF) [56] recognizes the D-box sequence (5′-RTTAYGTAAY-3′) [57] and connects circadian rhythm with metabolism. DBP regulates detoxification gene expression in the liver [57], TEF proteins contribute to muscle and pituitary development [58,59], and HLF is important for hematopoietic stem cell maintenance and has been implicated in leukemogenesis [60,61,62].

The CNC subfamily (NRF1, NRF2, NRF3, NF-E2, BACH1/2) [63] binds antioxidant response elements (ARE: 5′-TGACnnnGC-3′), usually in partnership with small MAF proteins [64]. NRF2–sMAF heterodimers activate antioxidant and detoxification genes [65], while NRF1 and NRF3 regulate proteostasis and cell proliferation [66,67]. BACH1 and BACH2 act as repressors by competing for ARE binding [68,69], contributing to heme homeostasis and immune regulation.

The MAF subfamily is divided into large MAFs (c-MAF, MAFA, MAFB, NRL) and small MAFs (MAFG, MAFK, MAFF) [64]. Large MAFs recognize MARE motifs (5′-TGCTGACTCAGCA-3′) [70,71] and regulate lineage-specific programs such as immune cell differentiation [72,73,74,75], insulin gene transcription in β-cells [76,77], and photoreceptor development [78]. Small MAFs lack transactivation domains and function mainly as dimerization partners for CNC proteins [64], thereby modulating oxidative stress responses and detoxification pathways.

The C/EBP subfamily (C/EBPα–ζ) recognizes CCAAT/enhancer sites [79,80] and regulates differentiation, metabolism, and inflammatory signaling [81,82,83,84]. Alternative translation generates isoforms (e.g., LAP/LIP of C/EBPβ) that fine-tune transcriptional output [85]. C/EBPα serves as a tumor suppressor in hematopoietic malignancies, whereas CHOP (C/EBPζ) mediates apoptosis during ER stress [86,87,88].

The CREB/ATF subfamily includes CREB1, CREM, ATF proteins, and the CREB3 family [89]. These proteins bind the CRE motif (5′-TGACGTCA-3′) and act as central regulators of stress response, proliferation, survival, and circadian rhythm [49,90,91]. CREB1 is activated by Ser133 phosphorylation, enabling the transcription of survival and growth-promoting genes [92,93]. CREM produces multiple isoforms with either activating or repressing functions [94,95,96], while ATF proteins respond rapidly to cellular stress to balance gene regulation [97]. Among these proteins, CREB3 (CREB3, CREB3L1–L4) and ATF6α/β contain type II transmembrane domains, which distinguish them from classical soluble nuclear CREB/ATF proteins [49,50].

A unifying feature of these TM-lacking bZIP proteins is their reliance on the LZ domain, which mediates homo- and hetero-dimerization. Homodimers reinforce binding specificity to canonical DNA motifs, while heterodimers expand binding repertoires and transcriptional diversity. For example, AP-1 complexes formed by Jun–Fos heterodimers strongly activate transcription [98], whereas Jun homodimers display weaker activity [99,100]. NRF2 requires heterodimerization with small MAFs to activate antioxidant genes [101], and CREB proteins often pair with ATF members to integrate CRE and TRE signaling [102]. These dimerization patterns provide the molecular basis for context-dependent and signal-specific transcriptional programs across the bZIP family.

### 3.3. bZIP Members Containing Transmembrane Domain

A subset of bZIP proteins contains type II TM domains, including CREB3 (CREB3L1–L4) and ATF6α/β [1]. These are anchored in the ER, Golgi, or INM. Stress triggers RIP by S1P/S2P, releasing their N-terminal bZIP domains to activate transcription [7,49]. The CREB3 family extends this paradigm by linking ER stress adaptation with chromatin tethering at the INM. ATF6 remains the classical ER stress sensor mediating unfolded protein response. Thus, TM-containing bZIP proteins are stress sensors that bridge membrane localization with gene regulation.

## 4. Nuclear Integrity Regulation

The NE not only defines the boundary of the genome but also maintains nuclear integrity by tethering chromatin to the INM. These tethering interactions provide a mechanical balance between outward pressure generated by chromatin compaction and inward forces transmitted by the nuclear lamina and the cytoskeleton [1,7]. Disruption of this balance compromises the nuclear stability, leading to NE rupture; chromatin leakage; and activation of RCD pathways, such as apoptosis [6] or the newly described karyoptosis [7].

Chromatin tethering mechanisms at the INM can be broadly divided into two categories. First, several classical INM proteins establish tethering through indirect interactions with chromatin, largely mediated by histone modifications or chromatin-associated adaptor proteins [6,103]. Representative examples include the LBR, which binds HP1 and H3K9-methylated histones, and LEM-domain proteins such as LAP2β, emerin, and MAN1, which associate with BAF and histone deacetylases [6,103]. PRR14 further exemplifies dynamic tethering through its interaction with HP1-bound heterochromatin [31]. These mechanisms stabilize heterochromatin positioning and nuclear shape, but they do not directly recognize genomic DNA.

Second, recent studies have identified a distinct group of INM-localized factors that tether chromatin through direct interaction with DNA. Among these factors, bZIP type II membrane-bound transcription factors such as CREB3 represent a novel paradigm [7]. The nucleoplasmic N-terminal domain of CREB3 binds to both lamins and chromatin DNA [7], thereby acting simultaneously as a structural tether and a transcriptional regulator. Under stress, RIP releases the transcriptionally active CREB3 fragment, disrupting chromatin anchorage and triggering nuclear instability that can culminate in karyoptosis [1,7].

Thus, nuclear integrity is maintained through a dual system of indirect chromatin tethering mediated by adaptor proteins and histone modifications, and direct DNA binding by transcription factors such as CREB3. This functional dichotomy underscores the complexity of nuclear architecture and highlights the emerging role of INM-anchored bZIP transcription factors in coupling chromatin dynamics with RCD.

### 4.1. Chromatin Tethering via Interaction with Chromatin

A major class of INM proteins tether chromatin not by directly binding to genomic DNA but through histone modifications or interactions with chromatin-associated proteins. These indirect tethering mechanisms are central to stabilizing heterochromatin at the nuclear periphery, reinforcing nuclear shape, and regulating gene expression in repressive domains.

One of the best-characterized factors is LBR. Instead of recognizing DNA directly, LBR anchors heterochromatin through binding to HP1 and recognizing H3K9-methylated histone tails via its Tudor domain, thereby positioning repressed chromatin at the NE [26,27]. Similarly, LEM-domain proteins such as LAP2β, emerin, and MAN1 mediate chromatin tethering through the adaptor protein BAF, which simultaneously binds DNA and nucleosomes [28,29]. LAP2β also interacts with HDAC3, promoting histone H4 deacetylation and facilitating the formation of transcriptionally repressive chromatin domains [34]. Emerin enhances this repression by directly activating HDAC3 enzymatic activity [35], while MAN1 exerts additional regulatory influence by binding Smad2/3, integrating chromatin organization with TGF-β signaling at the nuclear periphery [36].

Another example of a well-characterized factor is PRR14, which exhibits a cell cycle–regulated mechanism of tethering. PRR14 connects HP1-bound heterochromatin to lamin A, thereby stabilizing lamina–chromatin interactions during interphase and mitotic exit [31]. Unlike LBR or LEM-domain proteins, PRR14 functions dynamically, reflecting the adaptable nature of chromatin–lamina interactions during nuclear remodeling.

While lamina-associated domains (LADs) at the nuclear periphery are predominantly enriched in transcriptionally silent heterochromatin, the periphery is not exclusively repressive. Regions adjacent to the NPCs are frequently associated with actively transcribed genes, and inducible loci can be recruited to the NPCs in a regulated manner to facilitate efficient transcriptional activation [15,104,105]. This heterogeneity emphasizes that the nuclear periphery functions as a multifunctional compartment, integrating chromatin silencing through lamina-associated tethering with transcriptional activation at nuclear pores.

Collectively, these indirect tethering mechanisms highlight the reliance of most INM proteins on histone modifications and chromatin adaptor proteins rather than direct DNA binding. They ensure nuclear stability and the transcriptional silencing of heterochromatin domains, but they lack the ability to directly couple DNA to the nuclear lamina. This distinction sets the stage for understanding the unique contribution of bZIP type II transcription factors, such as CREB3, which tether chromatin through a direct DNA interaction (see Section 4.2).

### 4.2. Chromatin Tethering via Interaction with Genomic DNA in Chromatin

In contrast to indirect mechanisms that rely on histone modifications or chromatin-associated proteins, recent studies have identified a distinct set of INM proteins capable of tethering chromatin through direct binding to genomic DNA. This paradigm represents a fundamental shift in our understanding of nuclear integrity regulation, as it couples chromatin anchoring with transcriptional control.

The most prominent example of INM proteins is CREB3, a member of the bZIP type II membrane-bound transcription factor family. Unlike classical INM proteins, CREB3 contains a nucleoplasmic N-terminal domain that directly associates both lamins and chromatin DNA [7]. This dual tethering capacity allows CREB3 to balance mechanical forces within the nucleus: between the outward expansion force generated by compacted chromatin and cytoskeletal-linked outward pulling force via the LINC complex and fastening force, against the outward forces, of the INM via CREB3-mediated chromatin complex formation. By stabilizing this mechanical equilibrium, CREB3 maintains nuclear morphology and structural integrity under physiological conditions [7].

Beyond its canonical role in genetic information storage and transcriptional regulation, chromatin also serves as a critical mechanical element of the nucleoskeleton, contributing to nuclear shape, rigidity, and force transmission. This structural function is especially evident in the context of heterochromatin localized at the nuclear periphery. A notable example is the role of peripheral heterochromatin tethers in balancing mechanical stress across the NE. Poleshko et al. [106] demonstrated that tethering of chromatin to the nuclear lamina via LADs is essential for maintaining nuclear morphology and chromatin organization in differentiated cells. The disruption of these tethers results in chromatin detachment, nuclear rounding, and a loss of lineage-specific gene silencing, underscoring the structural contribution of chromatin to nuclear stability. Additionally, Stephens et al. [107] used micropipette aspiration and chromatin decondensation assays to reveal that chromatin acts as a viscoelastic scaffold within the nucleus. When chromatin was decondensed (e.g., with histone deacetylase inhibitors or DNase treatment), nuclei became mechanically softer and more deformable, whereas condensed chromatin increased nuclear stiffness and resistance to mechanical deformation. These findings position chromatin not merely as a passive genomic element but as an active mechanical component of the nucleoskeletal network, working in concert with lamins and LINC complexes to maintain nuclear integrity and mechanotransduction.

Importantly, CREB3 is not only a structural tether but also a signal-responsive transcription factor [108]. Under conditions of ER stress or other cellular insults, CREB3-FL undergoes RIP mediated by S1P and S2P [5]. This cleavage liberates the N-terminal bZIP transcriptional domain, which translocates to the nucleus to activate stress-response gene programs. However, this cleavage also disrupts CREB3-mediated chromatin tethering, thereby weakening nuclear stability. The excessive activation of this pathway results in NE rupture, chromatin leakage into the cytoplasm, and ultimately the induction of karyoptosis, a newly described form of RCD triggered by nuclear collapse [1,7].

Other members of the CREB3 subfamily (CREB3L1–L4) may share similar functions, as they also harbor type II transmembrane domains and bZIP motifs, although their precise roles at the INM remain less well characterized. One important report was published in 2021, indicating that CREB3L1 localizes to the NE and plays an important role in responding to the NE stress [109]. Moreover, immunocytofluorescence results indicate that ATF6α and ATF6β might localize to the NE [110]. More importantly, bZIP-containing membrane-bound transcription factors, including CREB3, CREB3L1, CREB3L3, CREB3L4, and ATF6α, were detected not only in the ER but also in the NE [4]. Together with CREB3-mediated chromatin tethering at the INM, although these proteins are classically known as an ER stressor sensor [50], it is possible to establish a hypothesis that these proteins exemplify how structural tethering and transcriptional regulation can converge in a single molecular system. Thus, direct tethering via bZIP type II transcription factors reveals a unique duality: these proteins act as mechanical stabilizers of nuclear integrity in their full-length form, while their proteolytically cleaved fragments function as transcriptional regulators of stress responses. This dual role distinguishes them from classical INM tethering proteins and positions them at the intersection of nuclear structure, genome regulation, and RCD.

### 4.3. The Molecular Mechanism of Karyoptosis Involving CREB3-CF

According to the canonical model, activation of CREB3 begins with its synthesis as a type II membrane integral protein. CREB3-FL is synthesized in the ER and integrated into the ER membrane [5]. In this configuration, the N-terminal domain faces the cytoplasm, while the C-terminal domain projects into the ER lumen, where it undergoes glycosylation and is retained. Upon specific signals or stimuli, CREB3-FL is transported to the Golgi complex via vesicular trafficking. There, it undergoes sequential proteolytic cleavage: the luminal domain is first processed by S1P, followed by RIP by S2P within the transmembrane domain [5]. The resulting cleaved fragment, CREB3-CF, is released into the cytoplasm and subsequently translocates to the nucleus through mechanisms that remain incompletely understood, where it functions as a transcription factor [5].

In contrast to this canonical ER–Golgi pathway, our data demonstrate that CREB3-FL is abundantly present in nuclear fractions and localizes to the INM, as confirmed by confocal microscopy [4,7]. At the nucleoplasmic side of the INM, the N-terminal bZIP domain of CREB3-FL interacts with chromatin DNA and the nuclear lamina composed of lamin A/C and lamin B [7]. This anchoring creates a fastening force at the nuclear periphery that counterbalances two opposing mechanical forces: the outward expansion force exerted by densely packed genomic DNA and the outward tensile force exerted by cytoskeletal tension transmitted through the LINC complex to maintain nuclear positioning [1,6,111]. The balance of these forces stabilizes nuclear architecture and preserves the maximally expanded spherical morphology of the nucleus. However, under conditions of cellular stress—such as ER stress induced by brefeldin A [112] or DNA damage triggered by UVB irradiation [7]—chromatin remodeling perturbs nuclear membrane tension [113]. This disturbance results in the widespread activation of S2P across the INM (accessed on 1 October 2025, Human Protein Atlas, https://www.proteinatlas.org/ENSG00000012174-MBTPS2/subcellular), and aberrant bulk cleavage events disrupt the force equilibrium at the NE.

The local cleavage of CREB3-FL at the INM is tightly regulated and allows transcriptional reprogramming in response to extracellular stimuli or growth factor signals. The release of CREB3-CF from the INM results in transient heterochromatin detachment, chromatin remodeling into euchromatin, and the subsequent activation of target gene expression. However, when the excessive or aberrant cleavage of CREB3-FL occurs under pathological stress, the mechanical balance at the INM is disrupted. This disequilibrium destabilizes the nuclear membrane, leading to rupture and ultimately inducing a distinct form of RCD termed karyoptosis.

Supporting this model, our findings show that the overexpression of CREB3-CF itself is sufficient to trigger karyoptosis. Ectopically expressed CREB3-CF interferes with the endogenous interactions between CREB3-FL, the nuclear lamina, and chromatin DNA at the nuclear periphery. This dominant-negative effect mimics the consequences of aberrant CREB3-FL cleavage, weakening chromatin–lamina attachments and reducing nuclear integrity. Thus, cells undergo NE rupture and karyoptotic cell death (Figure 3).

### 4.4. Potential Roles of Chromatin Acting as a Nucleoskeleton

Beyond its function as the carrier of genetic information, chromatin also plays a central role as a structural element of the nucleoskeleton [114]. By interacting with the INM and the nuclear lamina, chromatin provides mechanical counterforces that preserve nuclear shape and elasticity [114]. Compacted chromatin generates outward pressure, while the lamina and the LINC complex convey inward tensile forces from the cytoskeleton. The dynamic balance of these opposing forces is essential for nuclear integrity [111], and the disruption of chromatin–lamina attachments leads to nuclear rupture and RCD, including apoptosis and the recently described karyoptosis [7].

While many INM proteins tether chromatin indirectly through histone modifications or chromatin-binding adaptors, type II membrane-bound transcription factors such as CREB3 have recently emerged as direct chromatin tethers [1,7]. However, it is unlikely that CREB3 monomers alone could anchor and regulate the entire genome at the nuclear periphery. Instead, the ability of bZIP transcription factors to form both homodimers and heterodimers appears to be a critical determinant of their chromatin tethering capacity.

The bZIP domain combines a basic region for DNA binding with an LZ motif for dimerization, enabling the recognition of specific sequence motifs [43]. Although the canonical ACGT core is widely conserved, subtle variations define the specificity of different bZIP family members [39]. For instance, certain dimers preferentially recognize the CRE, others the TRE, as well as the ARE or the MARE. These variations allow dimerized bZIP proteins to discriminate among a broad spectrum of genomic targets [38,43].

Through this combinatorial DNA-binding code, CREB3 may extend its functional reach by partnering with other bZIP family members such as ATF6, CREB3L1–L4, or classical CREB/ATF proteins. Homodimers of CREB3 would reinforce binding to canonical CRE motifs, whereas heterodimers would broaden recognition to TRE- or ARE-like sequences, thus increasing the number of potential tethering sites across the genome [47]. Such versatility may allow bZIP dimers to organize genome-scale chromatin anchoring at the nuclear lamina, thereby positioning chromatin as an active nucleoskeletal element rather than a passive substrate. Stresses originated from ER by brefeldin A treatment and DNA damage response caused by UVB induced CREB3-FL cleavage-mediated CREB3-CF production and NE rupture via alterations in the nuclear membrane tension produced by chromatin remodeling [1,7,115]. Finally, dimerization dynamics add another regulatory layer. The N-terminal domains of CREB3 subfamily members form homodimers to reinforce DNA binding at CRE motifs [45,116], while heterodimerization with ATF or other bZIP partners broadens DNA recognition specificity and integrates diverse stress signals [47]. Such dimeric versatility fine-tunes the decision between stress adaptation and NE breakdown

We propose a model in which chromatin functions as a dynamic scaffold for dimerized bZIP type II transcription factors. In this framework, CREB3-FL and related proteins stabilize nuclear architecture by bridging lamins and chromatin DNA [7], while stress-induced proteolysis disrupts these tethers, weakening the NE and leading to karyoptosis [1]. This perspective highlights the duality of chromatin as both a genetic material and mechanical support, and underscores how dimerization-dependent DNA recognition by bZIP transcription factors integrates nuclear structure with RCD.

## 5. Future Directions

Although significant advances have been made in understanding the canonical ER-to-Golgi cleavage and transcriptional activation of type II membrane-bound transcription factors—such as members of the CREB3 and ATF6 families—their subcellular dynamics, nuclear functions, and non-canonical activation pathways remain incompletely understood. Traditionally regarded as the ER-localized sensors of cellular stress, recent evidence suggests that the full-length forms of CREB3 and certain CREB3L proteins may localize to the INM, where they participate in chromatin tethering, nuclear lamina interaction, and NE integrity maintenance under physiological conditions. Notably, the proteolytic machinery responsible for their activation—S1P and S2P—has been detected in compartments beyond the Golgi. Data from the Human Protein Atlas indicates that S2P is present in the nucleoplasm, raising the possibility that RIP of INM-localized transcription factors can occur within the nucleus itself, independent of Golgi trafficking. This nuclear-localized RIP mechanism has potential implications for stress-induced nuclear collapse, as observed in karyoptosis, and may represent an unrecognized regulatory axis in cell fate decisions.

Our unpublished results suggest that type II membrane-bound transcription factors indeed localize to the INM. Moreover, although the onset and duration of karyoptosis differ between cell types, CREB3 family members consistently induced this mode of cell death. Until recently, research on karyoptosis has been limited and rarely recognized. However, given that UVB irradiation induces karyoptosis with relatively high potency (16–40%), the ability to regulate this process may have broad implications for disease therapy and host defense.

Despite these advances, several testable hypotheses remain unproven. First, the precise subcellular distribution of CREB3, ATF6, and their cleaved fragments under diverse physiological and pathological conditions must be clarified. Second, it is critical to determine whether INM-localized pools of these proteins are functionally distinct from their ER- or Golgi-associated counterparts. Third, it remains to be elucidated how membrane-bound bZIP transcription factors contribute to nuclear architecture and how their cleavage initiates nuclear disassembly or chromatin detachment. Finally, the relationship between tissue-specific expression or localization of type II membrane-bound transcription factors and disease phenotypes—such as neurodegeneration, metabolic disorders, and cancer progression—requires systematic investigation.

To address these hypotheses, suitable experimental approaches include high-resolution live-cell imaging and subcellular fractionation to map dynamic protein localization, combined with protease activity assays to monitor RIP events at the INM. In parallel, functional genomics and proteomics will be essential to uncover the context-specific roles of full-length versus cleaved forms in different disease models. Mechanobiological assays and nuclear force-mapping techniques could further test the proposed model of S2P as a mechano-driven protease regulating INM integrity.

Understanding the dual roles of these transcription factors—as both transcriptional effectors and mechanical regulators at the nuclear periphery—will provide new insight into how cells coordinate stress signaling with structural integrity. This knowledge may ultimately facilitate the development of novel therapeutic strategies aimed at targeting CREB3/ATF6 signaling in diseases where nuclear membrane rupture, unresolved ER stress, or aberrant transcriptional responses play central roles.

## Figures and Tables

**Figure 1 ijms-26-10896-f001:**
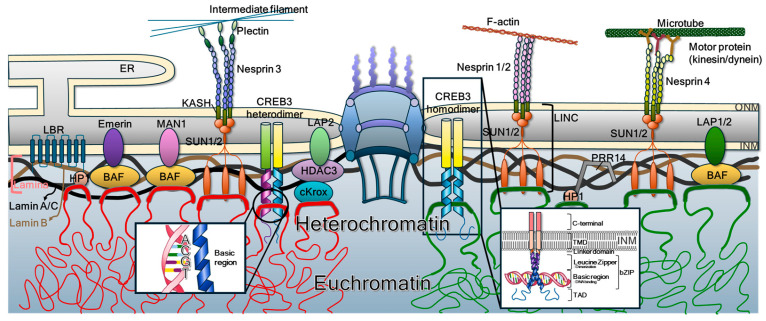
General structure and molecular constituents of the nuclear membrane. The ONM is continuous with the ER and is connected to the INM through the lipid bilayer surrounding the NPCs. Proteins destined for the INM traverse the NPCs or adjacent membrane regions to reach their final localization. The nucleoplasmic face of the INM is supported by the nuclear lamina, a dense fibrillar network composed primarily of lamin A/C and lamin B, which provides structural stability. Chromatin tethering at the INM is mediated by several integral proteins, including LBR, emerin, MAN1, SUN1/2, and LAP1/2, which interact indirectly with chromatin via adaptor proteins such as histones, HP1, BAF, and HDAC3. In addition, bZIP-containing type II membrane-bound transcription factors such as CREB3 can form homo- or heterodimers with related family members. These dimers directly bind chromatin DNA, revealing a previously unrecognized role for membrane-anchored transcription factors in maintaining nuclear structure and regulating gene expression. The nucleotide sequence, ACGT, is a canonically conserved ACGT core motif recognized by CREB3.

**Figure 2 ijms-26-10896-f002:**
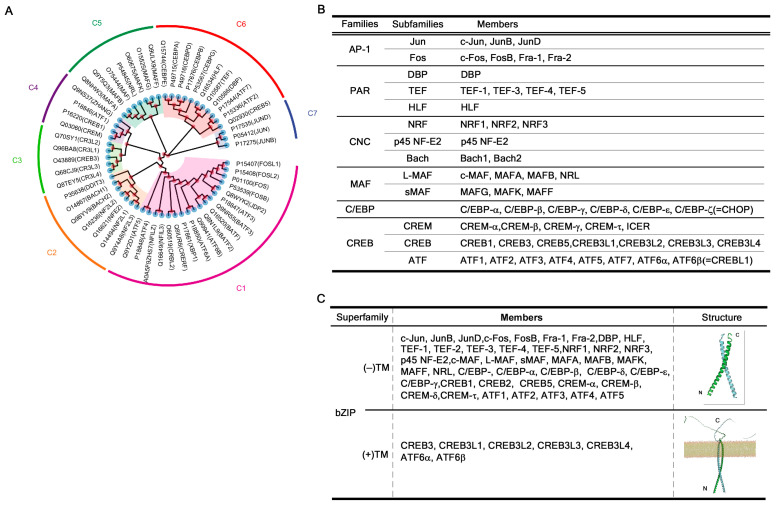
bLZ-containing type II membrane transcription factors. The bZIP superfamily of transcription factors encompasses 55 known genes in humans (**A**), all of which share a conserved bZIP domain that facilitates DNA binding and dimerization. These proteins are classified into seven major subfamily groups (**B**)—AP-1, PAR, CNC, MAF, C/EBP, and CREB—based on structural similarity of their bZIP domains, preference for specific DNA consensus motifs, dimerization specificity, target gene function, expression patterns across tissues and cell types, and phylogenetic relationships. Among these groups, type II membrane-bound transcription factors are found exclusively within the CREB subfamily. Specifically, CREB3, CREB3L1, CREB3L2, CREB3L3, and CREB3L4 from the CREB subfamily, along with ATF6α and ATF6β from the ATF subfamily, possess transmembrane domains and localize to membranous compartments (**C**). The classical activation mechanism of these proteins, primarily based on ER and Golgi localization, is illustrated in Figure 3A. However, recent studies have redefined the subcellular localization and functions of CREB3, revealing its presence at the INM and its role in a novel RCD pathway termed karyoptosis. These findings point to an emerging paradigm in which membrane-anchored transcription factors serve as key regulators of nuclear integrity and cellular fate. An evolutionary phylogenetic analysis further supports this classification: the bZIP family forms a rooted evolutionary tree using clusters C6+C7 as the outgroup, with the ingroup topology organized as (((C1, C2), (C3, C4)), C5). In this structure, C1 and C2 constitute sister groups, as do C3 and C4, while C5 branches as the sister clade to the combined C1–C4 lineage.

**Figure 3 ijms-26-10896-f003:**
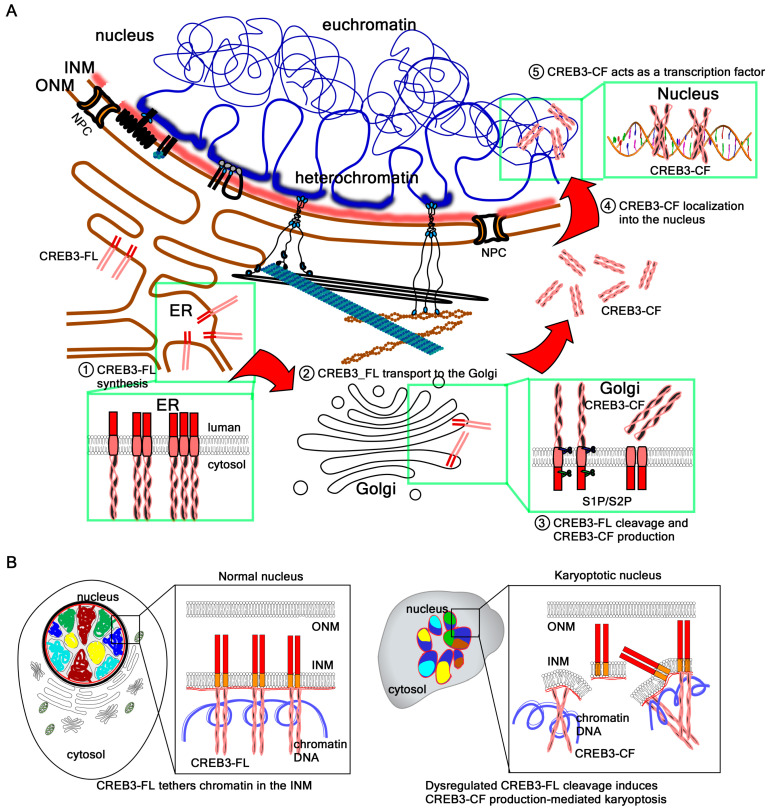
Equilibrium disturbance evokes karyoptosis. (**A**) The classical activation model of CREB3. The classical model of CREB3 activation, shared with other type II membrane transcription factors such as ATF6, suggests that full-length CREB3 is synthesized in the ER, trafficked to the Golgi apparatus, and cleaved by S1P and S2P. This cleavage releases the N-terminal cytosolic domain, which then translocates to the nucleus to function as an active transcription factor. The red arrows indicates sequential order of CREB3 processing. (**B**) New model of CREB3 cleavage in the INM. Recent studies have revised this paradigm. Emerging evidence demonstrates that CREB3 localizes directly to the INM via an as-yet-undefined mechanism following ER synthesis. At the INM, the N-terminal region of CREB3, located just before its transmembrane domain, interacts directly with chromatin DNA and nuclear lamins (lamin A/C and lamin B), anchoring chromatin to the nuclear periphery. This tethering is essential for maintaining mechanical equilibrium within the nucleus, balancing the outward expansive force of densely packed chromatin and the inward pulling forces mediated by the cytoskeleton through the nuclear lamina and LINC complex. Disruption of this tethering—such as through the aberrant cleavage of CREB3—compromises the structural integrity of the NE, resulting in chromatin detachment and catastrophic nuclear rupture. This process initiates a unique form of RCD termed karyoptosis, characterized by the explosive disassembly of the nucleus and chromatin leakage into the cytosol.

## Data Availability

No new data were created or analyzed in this study. Data sharing is not applicable to this article.

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
