# Peer review of "New Roles of bZIP-Containing Membrane-Bound Transcription Factors in Chromatin Tethering and Karyoptosis"

_ijms, 2025, doi:10.3390/ijms262210896_

Round 1
Reviewer 1 Report (Previous Reviewer 4)
Comments and Suggestions for Authors
The authors have significantly revised the initial version of the manuscript.
The main drawback of the work is that the experimental foundation of the review is based on a single publication by the authors. The major findings in this study (5) were obtained under artificial conditions where CREB3 was overexpressed in a cell culture. It is uncertain how these results can be generalized to all eukaryotic organisms and propose a novel paradigm in which “the nuclear envelope (NE) acts as a mechanosensitive signaling hub with implications for disease and treatment” (lines 18-19). The Review type of manuscripts should summarize the data of several other works, offering novel concepts.
Without a doubt, the concept proposed by the authors holds great interest for science. However, we need to wait at least for the publication of the new experimental data referenced by the authors (line 491). Another option is to publish this manuscript as a hypothesis rather than a review if the journal accepts publications in this format. Unfortunately, in the current form the manuscript cannot be considered sufficiently supported from a scientific perspective. The scientific standards of the IJMS journal do not allow for the publication of hypotheses that are not sufficiently supported by evidence.
Author Response
Comment. The authors have significantly revised the initial version of the manuscript. The main drawback of the work is that the experimental foundation of the review is based on a single publication by the authors. The major findings in this study (5) were obtained under artificial conditions where CREB3 was overexpressed in a cell culture. It is uncertain how these results can be generalized to all eukaryotic organisms and propose a novel paradigm in which “the nuclear envelope (NE) acts as a mechanosensitive signaling hub with implications for disease and treatment” (lines 18-19). The Review type of manuscripts should summarize the data of several other works, offering novel concepts. Without a doubt, the concept proposed by the authors holds great interest for science. However, we need to wait at least for the publication of the new experimental data referenced by the authors (line 491). Another option is to publish this manuscript as a hypothesis rather than a review if the journal accepts publications in this format. Unfortunately, in the current form the manuscript cannot be considered sufficiently supported from a scientific perspective. The scientific standards of the IJMS journal do not allow for the publication of hypotheses that are not sufficiently supported by evidence.
Response. We thank the reviewer for recognizing the importance of our revised manuscript and for the insightful comments regarding its scientific foundation. We fully understand and respect the reviewer’s concern that the current review may appear to rely heavily on our previously published study, particularly one that involves CREB3 overexpression in cultured cells.
However, we would like to clarify that the central concept presented in this review—that certain type II membrane-bound transcription factors such as CREB3 can localize to the inner nuclear membrane (INM), tether chromatin, and participate in nuclear membrane rupture and regulated cell death—is not solely based on artificial conditions. Instead, our model integrates a broad spectrum of evidence, including the following:
- Previously published data from other groups reporting INM localization of type II membrane transcription factors such as CREB3, ATF6, CREB3L1 (also known as OASIS), and CREB3L3. These observations were made independently, although their physiological relevance was not fully explored at the time.
- Endogenous CREB3 cleavage events observed in response to natural stimuli, such as ER stress. In fact, we and others have shown that agents like brefeldin A (BFA) can induce structural disruption between the outer and inner nuclear membranes, leading to CREB3 cleavage and nuclear envelope destabilization—phenomena confirmed by electron microscopy.
- Physiological relevance of CREB3-dTM: Alternative splicing generates CREB3 isoforms lacking transmembrane domains (CREB3-FL-dTM), which also points to naturally occurring regulatory forms. These isoforms further support the notion that CREB3 biology extends beyond overexpression models.
- Technical advancement and methodological value: One of the key contributions of our review lies in providing guidance on how to experimentally access and study membrane-bound transcription factors located in the INM. Many previous studies may have failed to detect these molecules due to inappropriate extraction protocols or insufficient attention to subnuclear localization. We believe this review will be instrumental for future investigations in the field.
We acknowledge that karyoptosis, as a newly proposed form of regulated cell death, remains in its early stages of characterization, and we agree that more studies will be needed to solidify its generalizability across eukaryotic systems. Nevertheless, the goal of this review is not to prematurely establish karyoptosis as a universally accepted paradigm but rather to compile and contextualize the emerging molecular insights that warrant future exploration. We have revised the manuscript accordingly to better frame our claims within the context of emerging evidence and to avoid overgeneralization. We invite the reviewer to refer to the newly highlighted sections in the revised manuscript for clarification. We also appreciate the reviewer’s suggestion that this manuscript might fit a “hypothesis” format. If the editorial board considers this a more appropriate classification, we are open to resubmitting the manuscript under that category, if such an option exists in IJMS.
Once again, we are grateful for the constructive feedback, which has helped us improve the clarity and scientific rigor of our work. Figures are contained in an attached PDF file.
Figure 1. Localization of CREB3 and ATF6 subfamily members to the nuclear membrane: evidence from previous studies (A) and confirmation by our group (B). Panel A summarizes earlier reports suggesting that members of the CREB3 subfamily (e.g., CREB3, CREB3L1, CREB3L2, CREB3L3, CREB3L4) and ATF6 subfamily (ATF6α, ATF6β) may localize to the nuclear membrane. Panel B shows our own experimental observations confirming the localization of these type II membrane-bound transcription factors to the inner nuclear membrane (INM), supporting their potential roles in chromatin tethering and nuclear membrane dynamics.
Figure 2. ER-stress-induced cleavage of CREB3-FL and nuclear envelope damage. (A) Previously published data showing that treatment with brefeldin A, an ER-stress-inducing agent, triggers proteolytic cleavage of full-length CREB3 (CREB3‑FL) and induces structural defects in the nuclear envelope. (B) Independent confirmation by our group demonstrating CREB3‑FL cleavage and nuclear envelope disruption following brefeldin A treatment. Panels B–F show transmission electron microscopy (TEM) images illustrating nuclear envelope deformation, membrane discontinuity, and chromatin leakage.

Reviewer 2 Report (New Reviewer)
Comments and Suggestions for Authors
Summary
This review highlights emerging roles of type II membrane-bound bZIP transcription factors, particularly the CREB3 family, in chromatin tethering, nuclear envelope (NE) integrity, and the recently described cell death mode termed karyoptosis. The topic is timely and relevant to researchers studying nuclear mechanics and regulated cell death. However, several areas require clarification and tighter organization to enhance the manuscript’s rigor and accessibility.
- Novelty and evidence balance
The review places strong emphasis on the authors’ recent findings regarding CREB3-mediated karyoptosis, while broader independent evidence remains limited. Interpretations about INM-localized cleavage and mechanoregulation would benefit from being clearly framed as emerging hypotheses with more cautious language. - Definition and positioning of karyoptosis
The concept should be more clearly differentiated from other nuclear-rupture-associated regulated cell death pathways (e.g., apoptosis, necroptosis, pyroptosis, ferroptosis, NETosis). Key features, biomarkers, and physiological relevance should be summarized to guide readers. - Supportive evidence for mechanistic claims
Some statements appear overstated or under-referenced. Additional citations supporting CREB3 localization, chromatin tethering functions, and relevance across cell types would strengthen the review. - Language and formatting
Several sections contain grammatical errors or inconsistent use of abbreviations. Editorial polishing would improve clarity and professionalism.
Author Response
Comment. This review highlights emerging roles of type II membrane-bound bZIP transcription factors, particularly the CREB3 family, in chromatin tethering, nuclear envelope (NE) integrity, and the recently described cell death mode termed karyoptosis. The topic is timely and relevant to researchers studying nuclear mechanics and regulated cell death. However, several areas require clarification and tighter organization to enhance the manuscript’s rigor and accessibility.
Response. We thank the reviewer’s recognition and consideration. We provide additional explanation, followed by the reviewer’s suggestions in a point-by-point manner below.
Comment 1. Novelty and evidence balance. The review places strong emphasis on the authors’ recent findings regarding CREB3-mediated karyoptosis, while broader independent evidence remains limited. Interpretations about INM-localized cleavage and mechanoregulation would benefit from being clearly framed as emerging hypotheses with more cautious language.
Response 1. We thank the reviewer for the invaluable and constructive comments. We agree that the manuscript should balance novelty with broader independent evidence and present current interpretations as emerging hypotheses rather than definitive conclusions.
As mentioned in our response to Reviewer #1, we have experimentally confirmed that karyoptotic events—such as nuclear deformation and nuclear envelope rupture triggered by brefeldin-A-induced ER stress—are observable phenomena and not artifacts of CREB3 overexpression. Moreover, the localization of CREB3 subfamily members and ATF6 at the nuclear membrane has been independently reported in previous studies, although the physiological significance of this localization was not appreciated at that time.
To address the reviewer’s comment and strengthen the conceptual framework, we expanded the section describing the role of chromatin as a structural element of the nucleoskeleton. Chromatin is not only a carrier of genetic information but also a mechanical component that contributes to nuclear stiffness, shape maintenance, and mechanotransduction. For example, Poleshko et al. (Cell, 2013) demonstrated that peripheral heterochromatin tethering to the nuclear lamina via lamina‑associated domains (LADs) is required to maintain nuclear morphology and gene silencing. Likewise, Stephens et al. (Mol Biol Cell, 2017) showed that chromatin compaction directly increases nuclear rigidity, while chromatin decondensation softens the nucleus independent of lamin levels. These studies support that chromatin acts as an internal scaffold—a nucleoskeleton—that counterbalances external cytoskeletal forces transmitted through the nuclear envelope.
In our review, we suggest—now explicitly framed as a hypothesis—that CREB3 at the inner nuclear membrane may function as a chromatin tethering factor, maintaining the equilibrium between peripheral chromatin forces and cytoskeleton-mediated tension. When aberrant cleavage releases CREB3 from the nuclear membrane, chromatin detaches from the INM, disturbing this force balance and ultimately leading to nuclear envelope rupture (karyoptosis). We have revised the manuscript to reflect that these concepts represent an emerging mechanistic model requiring further validation, and we adopted more cautious language accordingly. Revisions were made in the manuscript (highlighted text in Section 4.2).
Comment 2. Definition and positioning of karyoptosis. The concept should be more clearly differentiated from other nuclear-rupture-associated regulated cell death pathways (e.g., apoptosis, necroptosis, pyroptosis, ferroptosis, NETosis). Key features, biomarkers, and physiological relevance should be summarized to guide readers.
Response 2. We appreciate the reviewer’s insightful suggestion. Given that karyoptosis is a recently identified form of regulated cell death (RCD) characterized by nuclear envelope rupture and chromatin leakage, we agree it is important to contextualize this mechanism alongside other nuclear-rupture-associated RCDs. Although karyoptosis is still in its early conceptual stages, we have integrated a concise overview in the Introduction Section of the revised manuscript to help readers differentiate it from apoptosis, necroptosis, and other RCD pathways. This addition also references our prior review on UVB-induced cell death modalities (Chen et al., Free Radic. Res. 2024), which outlines distinct morphological and molecular features associated with each type of RCD, including karyoptosis. We hope this better defines the novelty and scope of the current work.
Comment 3. Supportive evidence for mechanistic claims. Some statements appear overstated or under-referenced. Additional citations supporting CREB3 localization, chromatin tethering functions, and relevance across cell types would strengthen the review.
Response 3. We sincerely apologize for any lack of citation and thank the reviewer for pointing out this important issue. In response, we have carefully re-evaluated the entire manuscript and added appropriate references wherever mechanistic claims regarding CREB3 localization, chromatin tethering, and functional relevance were previously under-supported.
We acknowledge that karyoptosis is a newly proposed form of regulated cell death and, therefore, the number of directly related studies is currently limited. However, extensive prior work has investigated CREB3 family members, and careful examination of these studies reveals indirect evidence supporting our claims. For example, CREB3 localization to the nuclear membrane has been observed in several confocal imaging studies, although the functional significance was not previously recognized (e.g., Mol Biol Cell. 2006; 17:413–426). In addition, CREB3L1 was recently shown to participate in nuclear envelope regeneration, particularly under DNA damage-induced nuclear stress, further supporting a potential role in nuclear membrane remodeling (Cell Death Discov. 2021; 7:152). While direct studies linking CREB3 to chromatin tethering are still emerging, we hope the reviewer will understand that this is an evolving area of research, and our interpretation is based on careful synthesis of available experimental observations.
We have now integrated these citations into the revised manuscript to reinforce our mechanistic discussion. Please refer to the highlighted changes in the revised manuscript.
Comment 4. Language and formatting. Several sections contain grammatical errors or inconsistent use of abbreviations. Editorial polishing would improve clarity and professionalism.
Response 4. We sincerely appreciate the reviewer’s helpful comment regarding language and formatting. In response, we have carefully reviewed the entire revised manuscript to correct grammatical errors, improve the sentence clarity, and ensure consistent use of abbreviations throughout the text. Additionally, we utilized the professional English editing service provided by MDPI Author Services to enhance the overall readability and presentation of the manuscript. All relevant corrections and edits are highlighted in the revised version for your reference.

Round 2
Reviewer 1 Report (Previous Reviewer 4)
Comments and Suggestions for Authors
This work proposes a novel and interesting concept of nuclear membrane mechanosensitivity and the role of bZIP factors in this phenomenon. The authors present a body of experimental data that serves as the foundation for their model. Currently, these data can only be considered as potential indications and a starting point for further research. The model itself requires substantial additional experimental validation and development.
Consequently, this work cannot be accepted as a Review of existing data, but could be considered for publication as a Hypothesis, Perspective, or similar paper type. If the journal editor deems such an option feasible, the manuscript may be recommended for publication.
Author Response
[Reviewer 1-Round2]
Comment. This work proposes a novel and interesting concept of nuclear membrane mechanosensitivity and the role of bZIP factors in this phenomenon. The authors present a body of experimental data that serves as the foundation for their model. Currently, these data can only be considered as potential indications and a starting point for further research. The model itself requires substantial additional experimental validation and development. Consequently, this work cannot be accepted as a Review of existing data, but could be considered for publication as a Hypothesis, Perspective, or similar paper type. If the journal editor deems such an option feasible, the manuscript may be recommended for publication.
Response. We sincerely thank the reviewer for their thoughtful evaluation and for the constructive suggestion to consider reclassifying the manuscript as a Hypothesis article rather than a traditional Review. We fully agree with the reviewer’s perspective that, while our model is grounded in emerging experimental observations, further validation is indeed necessary to establish it as a fully substantiated mechanism. Accordingly, we have revised and resubmitted the manuscript under the Hypothesis article type (manuscript ID: ijms-3947452R2), and we kindly request the reviewer’s consideration of the revised version within this framework.
Round 3
Reviewer 1 Report (Previous Reviewer 4)
Comments and Suggestions for Authors
In its present form, the manuscript can be recommended for publication.
This manuscript is a resubmission of an earlier submission. The following is a list of the peer review reports and author responses from that submission.
Round 1
Reviewer 1 Report
Comments and Suggestions for Authors
This review provides a comprehensive overview of the roles of the nuclear envelope, and specifically the inner nuclear membrane, as a dynamic signaling hub. The topic is of high interest to the fields of cell biology, cancer biology, and mechanotransduction.
The manuscript is well-written and logically structured. My only major recommendation fo authors is to condense Section 3 (bZIP family members). This section is exceptionally detailed but feels disproportionate to the main focus of the review.
Minor Points:
- Line 388 - wrong citation format
- Lines 283 and 261 - conflicting numbers of bZIP genes
Author Response
Overall comment 1. This review provides a comprehensive overview of the roles of the nuclear envelope, and specifically the inner nuclear membrane, as a dynamic signaling hub. The topic is of high interest to the fields of cell biology, cancer biology, and mechanotransduction. The manuscript is well-written and logically structured. My only major recommendation for authors is to condense Section 3 (bZIP family members). This section is exceptionally detailed but feels disproportionate to the main focus of the review.
Response to overall comment. We sincerely thank the reviewer for recognizing the novelty and importance of our review and for the constructive feedback. We carefully revised the manuscript by incorporating the valuable suggestions from all four reviewers. The minor comments raised have also been addressed, and all revisions are summarized in a point-by-point response below. We kindly ask the reviewer to evaluate the revised version once again.
Minor Points:
Comment 1. Line 388 - wrong citation format
Response 1. We apologize for the oversight in the citation format. This error has been corrected in the revised manuscript, and we carefully re-checked the entire reference list to ensure consistency and accuracy.
Comment 2. Lines 283 and 261 - conflicting numbers of bZIP genes
Response 2. We thank the reviewer for this careful comment. In response to the suggestion, we re-examined the UniProt database (https://www.uniprot.org/uniprotkb?query=%28family%3A%22bZIP+family%22%29&facets=reviewed%3Atrue%2Cmodel_organism%3A9606). Our analysis indicated that the previously cited number of 53 bZIP transcription factors was outdated. Accordingly, we have updated the revised manuscript to reflect the current value of 55 bZIP genes.
Reviewer 2 Report
Comments and Suggestions for Authors
The review “The nuclear skeleton rewired: Inner nuclear membrane transcription factors in chromatin tethering and karyoptosis” claims to present the insights into the nuclear skeleton and nuclear membrane as a mechanosensitive signaling hubs, and linking these nuclear parts to the transcriptional regulation, homeostasis and cell fate (death). The overall problem is very interesting and deserves the attention. However, the review is not well structured. Part 2 is too long and describes the general information, and Part 3, except for Section 3.7, has little to do with the topic of the review. The whole text contains a lot of repetitions, often verbatim. The main idea of the review is concentrated only in Part 4, and specifically, in Section 4.2. I propose to reformat the text, to shorten Sections 2 and 3 to a minimal information, which is important for the understanding of the ideas highlighted in the title. Or the authors should consider renaming the review, because its content does not correspond to the title. Another remark concerns the frequently mentioned (and put in the title) new mechanism of cell death—karyoptosis. I would prefer to have more information about its molecular mechanisms, not just its manifestations, and how they may be related to disruptions of the nuclear skeleton and the transcription factors associated with the inner nuclear membrane. I believe the review needs to be thoroughly reformatted.
Author Response
Overall comment. The review “The nuclear skeleton rewired: Inner nuclear membrane transcription factors in chromatin tethering and karyoptosis” claims to present the insights into the nuclear skeleton and nuclear membrane as a mechanosensitive signaling hubs, and linking these nuclear parts to the transcriptional regulation, homeostasis and cell fate (death). The overall problem is very interesting and deserves the attention. However, the review is not well structured. Part 2 is too long and describes the general information, and Part 3, except for Section 3.7, has little to do with the topic of the review. The whole text contains a lot of repetitions, often verbatim.
Response to the overall comment. We sincerely thank the reviewer for the constructive feedback and for recognizing the importance of the topic addressed in our review. Following the reviewer’s advice, we substantially shortened Part 2 by condensing general background information and reorganized the manuscript to reduce redundancy. We also strengthened the central arguments to ensure that Part 3 is more directly connected to the main theme. As a result, large portions of the manuscript were rewritten for clarity and focus. All revised and newly organized sections are highlighted in the revised version for ease of review.
Comment 1. The main idea of the review is concentrated only in Part 4, and specifically, in Section 4.2. I propose to reformat the text, to shorten Sections 2 and 3 to a minimal information, which is important for the understanding of the ideas highlighted in the title. Or the authors should consider renaming the review, because its content does not correspond to the title.
Response 1. We appreciate the reviewer’s thoughtful suggestion and fully agree with this point. In the revised manuscript, Sections 2 and 3 were substantially shortened to retain only the essential background information, while Section 4 was reorganized to emphasize our central perspective more clearly. In addition, we modified the title to “New roles of bZIP-containing transcription factors in chromatin tethering and karyoptosis” to better reflect the focus and content of the review. All changes and rearrangements are highlighted in the revised version for the reviewer’s reference.
Comment 2. Another remark concerns the frequently mentioned (and put in the title) new mechanism of cell death—karyoptosis. I would prefer to have more information about its molecular mechanisms, not just its manifestations, and how they may be related to disruptions of the nuclear skeleton and the transcription factors associated with the inner nuclear membrane. I believe the review needs to be thoroughly reformatted.
Response 2. We thank the reviewer for the constructive comments and fully agree with this point. In the revised manuscript, we have expanded the discussion of the molecular mechanisms underlying karyoptosis, particularly the role of CREB3-CF, which is now described in detail in Subsection 4.3. Furthermore, we added a new discussion in Subsection 4.4 addressing the emerging role of bZIP-containing transcription factors in both preserving genetic information through chromatin tethering and providing mechanical support to the nucleus.
Reviewer 3 Report
Comments and Suggestions for Authors
This manuscript presents a comprehensive review of bZIP transcription factors, with particular emphasis on type II membrane-bound members (CREB3 and ATF6) and their emerging roles in nuclear integrity and regulated cell death. The topic is timely and scientifically relevant, bridging classical bZIP biology with novel findings on INM localization and karyoptosis. The work demonstrates originality and potential impact for both basic research and translational studies. However, substantial revisions are necessary to improve clarity, organization, and the robustness of mechanistic claims.
Major concerns:
- The manuscript is dense and occasionally repetitive, particularly in sections 3 and 4.
- Certain mechanistic explanations (e.g., nuclear-localized RIP, karyoptosis) are complex and would benefit from schematic diagrams and simplified conceptual descriptions (optional).
- Subsections could be more concise, with clear topic sentences highlighting key points.
- Some claims, particularly regarding CREB3 INM localization and direct chromatin tethering, are supported by limited studies. Please clarify which conclusions are firmly established and which remain emerging hypotheses.
- Consider including additional references supporting recent findings in nuclear envelope mechanics, chromatin tethering, and regulated cell death.
- Further elaboration is needed on how membrane-bound bZIP factors integrate mechanical stress with transcriptional activity.
- Distinguish clearly between classical ER-to-Golgi pathways and potential INM-localized cleavage events.
- The “Future Directions” section is relevant, but could more explicitly highlight testable hypotheses, outline suitable experimental approaches, and point to key unresolved questions.
- Figure 1 is visually complex, but several elements are insufficiently explained in the legend. For example, the boxed region labeled “chromatin” depicts four nucleosomes and highlights Histone H3, yet this is not described in the caption. Moreover, chromatin is represented as an amorphous tangle of lines, which does not adequately reflect higher-order chromosome organization. A clearer representation could include the depiction of TADs, specification of nucleotide composition at the nuclear periphery, or annotation of heterochromatin versus euchromatin. Indicating nuclear proteins associated with chromatin at the INM interface would also improve accuracy. Overall, revising both the schematic and the legend to be less discursive and more descriptive would substantially enhance clarity and explanatory value.
- In the Figure 3, nuclear DNA is depicted only as generic heterochromatin and euchromatin. There is no reference to chromosome territories or their radial organization within the nucleus, which is known to be influenced by nucleotide composition and chromatin compaction. It would be highly valuable to depict nuclear DNA in relation to chromosome territories and to highlight their spatial relationship to nuclear pores. Such refinements would provide a more accurate and educational representation of nuclear architecture.
Minor concerns:
- Ensure all abbreviations are defined at first use.
- Consider shortening overly detailed descriptions of DNA motifs where not directly relevant to nuclear integrity discussion.
Author Response
Overall comment. This manuscript presents a comprehensive review of bZIP transcription factors, with particular emphasis on type II membrane-bound members (CREB3 and ATF6) and their emerging roles in nuclear integrity and regulated cell death. The topic is timely and scientifically relevant, bridging classical bZIP biology with novel findings on INM localization and karyoptosis. The work demonstrates originality and potential impact for both basic research and translational studies. However, substantial revisions are necessary to improve clarity, organization, and the robustness of mechanistic claims.
Response to the overall comment. We sincerely thank the reviewer for recognizing the novelty and significance of our work. Following the reviewer’s advice, we have substantially revised the manuscript to improve clarity, organization, and the robustness of mechanistic descriptions. We believe that these changes strengthen the overall impact of the review, and we kindly invite the reviewer to examine the revised version.
Major concerns:
Comment 1. The manuscript is dense and occasionally repetitive, particularly in sections 3 and 4.
Response 1. We thank the reviewer for this important observation. In the revised manuscript, Sections 3 and 4 have been thoroughly rewritten and reorganized to improve readability. Redundant statements were removed to avoid repetition, and the overall flow was streamlined. All revisions are highlighted in the revised manuscript.
Comment 2. Certain mechanistic explanations (e.g., nuclear-localized RIP, karyoptosis) are complex and would benefit from schematic diagrams and simplified conceptual descriptions (optional).
Response 2. We appreciate the reviewer’s constructive suggestion. To facilitate readers’ understanding, we expanded the description of the molecular mechanism of karyoptosis induced by CREB3-CF in Subsection 4.3. In addition, we have included a schematic diagram (Figure 3) to provide a simplified conceptual illustration of these mechanisms. All changes are available in the revised manuscript.
Comment 3. Subsections could be more concise, with clear topic sentences highlighting key points.
Response 3. We appreciate the reviewer’s helpful suggestion. In the revised manuscript, we have condensed the general background information and refined each subsection to be more concise, beginning with clearer topic sentences that highlight the key points. We also emphasized the dual roles of chromatin as both genetic material and mechanical support, which now provides a smoother transition to the discussion of future directions.
Comment 4. Some claims, particularly regarding CREB3 INM localization and direct chromatin tethering, are supported by limited studies. Please clarify which conclusions are firmly established and which remain emerging hypotheses.
Response 4. We appreciate the reviewer’s thoughtful suggestion. As karyoptosis is a newly identified form of regulated cell death, the current literature on CREB3 localization at the INM and its role in direct chromatin tethering remains limited. To address this, we have clarified in the revised manuscript which aspects are supported by established findings and which represent emerging hypotheses. In addition, we noted that three related research articles, prepared in collaboration with Dr. M. Fanto (King’s College London), are currently under review. To aid readers’ understanding, we also expanded the discussion in Subsection 4.4 on how bZIP-containing transcription factors may tether chromatin at the INM.
Comment 5. Consider including additional references supporting recent findings in nuclear envelope mechanics, chromatin tethering, and regulated cell death.
Response 5. We thank the reviewer for this valuable suggestion. In the revised manuscript, we have included additional references to strengthen the discussion of recent advances in nuclear envelope mechanics, chromatin tethering, and regulated cell death. While some of our own related studies are currently under review or in revision (e.g., demonstrating nuclear localization of S1P/S2P proteases, the mechanistic basis of CREB3 cleavage at the INM, and the efficacy of karyoptosis in cancer and neurodegenerative disease models), we have refrained from citing unpublished data. Instead, we expanded the reference list with published studies from the field to provide readers with broader context and supporting evidence. We kindly ask the reviewer’s understanding that our ongoing work will be available in future publications.
Comment 6. Further elaboration is needed on how membrane-bound bZIP factors integrate mechanical stress with transcriptional activity.
Response 6. We thank the reviewer for this invaluable question. Our central claim is that bZIP transcription factors with transmembrane domains, such as CREB3, represent a novel class of regulators that integrate mechanical stress with transcriptional activity. These factors tether chromatin to the INM through direct DNA binding, thereby coupling nuclear architecture to gene regulation. The nuclear envelope is highly dynamic and responds to diverse stimuli, including ER stress, DNA damage, cell cycle progression, and cell migration. In particular, stressors such as UVB irradiation that induce ER stress and DNA damage responses can trigger CREB3 cleavage, leading to detethering of chromatin and activation of transcriptional programs or cell death pathways. To further clarify this point, we expanded the discussion in Subsection 4.4, emphasizing the concept of chromatin acting as part of the nucleoskeleton and describing how membrane-bound bZIP transcription factors integrate structural stress with transcriptional outcomes.
Comment 7. Distinguish clearly between classical ER-to-Golgi pathways and potential INM-localized cleavage events.
Response 7. We appreciate the reviewer’s constructive comment. In the revised manuscript, we have clarified the distinction between the classical ER–Golgi activation pathway and the potential INM-localized cleavage events. Specifically, in Subsection 4.3 we expanded the discussion of CREB3-CF–mediated karyoptosis, describing that CREB3-FL cleavage at the INM may also be mediated by S1P and S2P. This notion is supported by our immunocytochemistry data for endogenous and exogenous S1P and S2P, which are currently under review in the Journal of Advanced Research. Furthermore, in Subsection 4.4 we propose that S2P activity at the INM may operate as a mechano-driven protease, providing a mechanistic basis for stress-induced chromatin detethering and nuclear rupture.
Comment 8. The “Future Directions” section is relevant, but could more explicitly highlight testable hypotheses, outline suitable experimental approaches, and point to key unresolved questions.
Response 8. We appreciate the reviewer’s invaluable suggestion. In the revised manuscript, we expanded the Future Directions section to more explicitly highlight testable hypotheses, outline suitable experimental approaches, and identify key unresolved questions. In addition, based on our preliminary unpublished results, we emphasized the potential significance of karyoptosis research in linking nuclear membrane integrity with regulated cell death. These revisions are now reflected in Section 5 of the manuscript.
Comment 9. Figure 1 is visually complex, but several elements are insufficiently explained in the legend. For example, the boxed region labeled “chromatin” depicts four nucleosomes and highlights Histone H3, yet this is not described in the caption. Moreover, chromatin is represented as an amorphous tangle of lines, which does not adequately reflect higher-order chromosome organization. A clearer representation could include the depiction of TADs, specification of nucleotide composition at the nuclear periphery, or annotation of heterochromatin versus euchromatin. Indicating nuclear proteins associated with chromatin at the INM interface would also improve accuracy. Overall, revising both the schematic and the legend to be less discursive and more descriptive would substantially enhance clarity and explanatory value.
Response 9. We apologize for the oversight and thank the reviewer for this helpful suggestion. In the revised manuscript, Figure 1 has been refined to more clearly depict nuclear architecture. Specifically, heterochromatin and euchromatin are now distinguished by line thickness, and CREB3 homo- and heterodimers are indicated at the INM. The structural representation of transmembrane bZIP factors such as CREB3 was improved in the boxed region, and color-coded chromatin surrounding NPCs was added to represent chromosome territories, consistent with Figure 3. To better reflect higher-order organization, the amorphous tangle of chromatin was replaced with a more ordered depiction of euchromatin. The figure legend was also rewritten to provide precise and descriptive annotations corresponding to each element.
Comment 10. In the Figure 3, nuclear DNA is depicted only as generic heterochromatin and euchromatin. There is no reference to chromosome territories or their radial organization within the nucleus, which is known to be influenced by nucleotide composition and chromatin compaction. It would be highly valuable to depict nuclear DNA in relation to chromosome territories and to highlight their spatial relationship to nuclear pores. Such refinements would provide a more accurate and educational representation of nuclear architecture.
Response 10. We appreciate the reviewer’s constructive comment. Following the suggestion, we refined Figure 3 to provide a more accurate representation of nuclear architecture. Specifically, we added nuclear pore complexes (NPCs) with their spatial organization related to nuclear pores in panel A and incorporated chromosome territories in panel B to illustrate changes during the karyoptosis process. The revised figure is now included in the manuscript.
Minor concerns:
Comment 1. Ensure all abbreviations are defined at first use.
Response 1. We apologize for the oversight. We carefully reviewed the entire manuscript and ensured that all abbreviations are defined at their first appearance.
Comment 2. Consider shortening overly detailed descriptions of DNA motifs where not directly relevant to nuclear integrity discussion.
Response 2. We appreciate the reviewer’s helpful comment. In the revised manuscript, Sections 3.2 and 3.3 were rewritten to shorten overly detailed descriptions of DNA motifs and to retain only content relevant to nuclear integrity. We also reorganized these sections to improve clarity and readability in line with the reviewer’s suggestion.
Reviewer 4 Report
Comments and Suggestions for Authors
The paper proposes a new paradigm of the nuclear envelope as an active regulatory interface where mechanically anchored transcription factors coordinate genome organization with stress signaling and cell death, advancing the understanding of nuclear architecture and function in health and disease.
Recent discovery highlights that bZIP transcription factor CREB3 localizes to the inner nuclear membrane (INM), where it tethers chromatin and serves as a stress sensor. Under stress, it undergoes regulated intramembrane proteolysis (RIP) by proteases S1P and S2P, which may also act at the INM, releasing active transcription factor that drives stress adaptation or cell death pathways. A novel form of regulated cell death termed karyoptosis is triggered when cleavage of CREB3 disrupts chromatin anchoring at the INM, causing nuclear rupture and loss of nuclear integrity due to mechanical imbalance.
There are several serious points that can be made about the manuscript.
- The authors use a nuclear skeleton model that is now outdated. According to current concepts of nuclear structure, there is no internal network similar to the cytoskeleton, and the only rigid architectural element is the nuclear envelope.
- The authors incorrectly describe the periphery of the nucleus as a region of inactive chromatin (paragraph 4.1). Areas near nuclear pores contain actively transcribed genes, and the process of recruiting genes to pores is regulated.
- Most part of the manuscript focuses on the description of proteins and processes that are unrelated to its title. It is not clear why a detailed overview of the bZIP family is presented, if only a few members of this family can bind to the membrane and participate in the processes discussed in the paper. Most of this information may be removed from the manuscript without affecting the main idea of the paper in any significant way.
- The main point of the review is a summary of the authors' own work on this topic (7). At the same time, the authors have already published a review in which they discussed their work (8).
Thus, the manuscript contains outdated and inaccurate concepts, and for the most part consists of irrelevant information. The work claims to offer a new perspective on the functions of the nuclear membrane, but the source of this viewpoint is only one experimental article by the authors that has not been sufficiently developed in other works. Clearly, the concept proposed by the authors cannot be considered sufficiently substantiated at this time from a scientific perspective.
Author Response
Overall comment. The paper proposes a new paradigm of the nuclear envelope as an active regulatory interface where mechanically anchored transcription factors coordinate genome organization with stress signaling and cell death, advancing the understanding of nuclear architecture and function in health and disease. Recent discovery highlights that bZIP transcription factor CREB3 localizes to the inner nuclear membrane (INM), where it tethers chromatin and serves as a stress sensor. Under stress, it undergoes regulated intramembrane proteolysis (RIP) by proteases S1P and S2P, which may also act at the INM, releasing active transcription factor that drives stress adaptation or cell death pathways. A novel form of regulated cell death termed karyoptosis is triggered when cleavage of CREB3 disrupts chromatin anchoring at the INM, causing nuclear rupture and loss of nuclear integrity due to mechanical imbalance. There are several serious points that can be made about the manuscript.
Response to overall comment. We sincerely thank the reviewer for the thoughtful summary and recognition of our work. In the revised manuscript, we have substantially reorganized and refined the text to improve clarity and focus. We carefully addressed all of the reviewer’s constructive comments and suggestions point by point, and the corresponding changes are highlighted throughout the revised version for ease of review.
Comment 1. The authors use a nuclear skeleton model that is now outdated. According to current concepts of nuclear structure, there is no internal network similar to the cytoskeleton, and the only rigid architectural element is the nuclear envelope.
Response 1. We appreciate the reviewer’s valuable comment and agree with this point. Following the reviewer’s guidance, we rewrote the second paragraph of the Introduction to avoid presenting an outdated “nuclear skeleton” model. The revised text now emphasizes the nuclear envelope and its associated complexes as the principal architectural scaffold, while describing chromatin as a dynamic component that contributes to nuclear membrane remodeling and mechanosensitive signaling. We believe this change provides greater clarity and aligns the manuscript with current concepts of nuclear structure.
Comment 2. The authors incorrectly describe the periphery of the nucleus as a region of inactive chromatin (paragraph 4.1). Areas near nuclear pores contain actively transcribed genes, and the process of recruiting genes to pores is regulated.
Response 2. We thank the reviewer for this careful comment. We agree that our original description of the nuclear periphery was oversimplified. In the revised manuscript, we substantially revised Sections 3 and 4 to improve clarity and incorporated the reviewer’s suggestion. Specifically, in Subsection 4.1 we now state: “While LADs at the nuclear periphery are predominantly enriched in transcriptionally silent heterochromatin, the periphery is not exclusively repressive. Regions adjacent to nuclear pore complexes (NPCs) are frequently associated with actively transcribed genes, and inducible loci can be recruited to NPCs in a regulated manner to facilitate efficient transcriptional activation (13, 104, 105). This heterogeneity emphasizes that the nuclear periphery functions as a multifunctional compartment, integrating chromatin silencing through lamina-associated tethering with transcriptional activation at nuclear pores.”
Comment 3. Most part of the manuscript focuses on the description of proteins and processes that are unrelated to its title. It is not clear why a detailed overview of the bZIP family is presented, if only a few members of this family can bind to the membrane and participate in the processes discussed in the paper. Most of this information may be removed from the manuscript without affecting the main idea of the paper in any significant way.
Response 3. We thank the reviewer for this thoughtful comment. We agree that a detailed overview of the entire bZIP family could appear disproportionate given that only a subset of these factors is membrane-bound and directly relevant to the focus of our review. In the revised manuscript, we therefore condensed the general description of the bZIP family and emphasized instead the subset of type II membrane-bound members, such as CREB3 and ATF6, that are most directly implicated in nuclear integrity and karyoptosis.
Nevertheless, we retained a concise overview of the broader bZIP family because recent findings suggest that combinatorial dimerization among bZIP proteins significantly expands their functional diversity (eLife. 2017;6:e19272). This study predicted that 53 human bZIP transcription factors could form as many as 1,431 potential dimers, including 22 possible homodimers. Our own unpublished results further support this notion by showing that CREB3 can form multimers with other CREB3 and ATF6 subfamily members, as confirmed by FRET, mammalian two-hybrid assays, and immunoprecipitation. These dimerization properties are directly relevant to our central argument, since homo- and heterodimer formation may broaden DNA-binding specificity and mechanistic functions at the nuclear periphery.
Thus, in the revised manuscript we have streamlined Sections 4.1, 4.2, and 4.4. focusing on how dimerization contributes to chromatin tethering and nuclear integrity, while removing unrelated details. We believe this provides a clearer rationale for why the bZIP family context is important for understanding the unique contributions of membrane-bound transcription factors such as CREB3.
Comment 4. The main point of the review is a summary of the authors' own work on this topic (7). At the same time, the authors have already published a review in which they discussed their work (8). Thus, the manuscript contains outdated and inaccurate concepts, and for the most part consists of irrelevant information. The work claims to offer a new perspective on the functions of the nuclear membrane, but the source of this viewpoint is only one experimental article by the authors that has not been sufficiently developed in other works. Clearly, the concept proposed by the authors cannot be considered sufficiently substantiated at this time from a scientific perspective.
Response 3. We thank the reviewer for this important comment. We agree that karyoptosis is a recently identified phenomenon and that the number of published references on this subject remains limited. While our laboratory has a long history of studying bZIP transcription factors, most previous work—both by our group and others—has emphasized their classical role as transcriptional regulators. Our earlier review (2024) focused specifically on UVB-induced CREB3 cleavage and its association with karyoptosis. In contrast, the present manuscript extends beyond that perspective by proposing a novel function for bZIP-containing membrane-bound transcription factors: their ability to tether chromatin to the INM through homo- and heterodimerization.
This new viewpoint is based not only on our prior publication but also on evidence supporting the combinatorial dimerization capability of bZIP transcription factors and the complexity of their DNA-binding landscapes (eLife. 2017;6:e19272). In addition, it builds upon ongoing studies from our laboratory, which are currently under peer review or revision. These results suggest that chromatin, in addition to preserving genetic information, may act as a structural nucleoskeleton that integrates nuclear mechanics with stress signaling. Although we acknowledge that this concept is still emerging, we believe it provides a valuable framework to stimulate further investigation in the field. We respectfully ask the reviewer to view this manuscript as an effort to articulate and consolidate a new direction of research, which we anticipate will be further substantiated as additional data become available.
Round 2
Reviewer 2 Report
Comments and Suggestions for Authors
No more suggestions.
Reviewer 3 Report
Comments and Suggestions for Authors
Thank you for thoroughly addressing the suggestions provided. I have no additional remarks.
Reviewer 4 Report
Comments and Suggestions for Authors
The authors have significantly revised the manuscript and considered the comments provided.
However, the most significant point is that the experimental foundation of this review is based on a single publication by the authors. The major findings in this study (5) were obtained under artificial conditions where CREB3 was overexpressed in a cell culture. It is uncertain how these results can be generalized to all eukaryotic organisms and propose a novel paradigm in which “the nuclear envelope (NE) acts as a mechanosensitive signaling hub with implications for disease and treatment”.
Without a doubt, the concept proposed by the authors holds great interest for science. However, we need to wait for the publication of the new experimental data referenced by the authors. Another option is to publish this manuscript as a hypothesis rather than a review if the journal accepts publications in this format. Unfortunately, in the current form the manuscript cannot be considered sufficiently supported from a scientific perspective.